# T-Cell Engager Therapy in Prostate Cancer: Molecular Insights into a New Frontier in Immunotherapy

**DOI:** 10.3390/cancers17111820

**Published:** 2025-05-29

**Authors:** Whi-An Kwon, Jae Young Joung

**Affiliations:** 1Department of Urology, Hanyang University College of Medicine, Myongji Hospital, Goyang 10475, Republic of Korea; 2Department of Urology, Center for Urological Cancer, National Cancer Center, Goyang 10408, Republic of Korea

**Keywords:** prostate cancer, T-cell engager (TCE), immunotherapy, PSMA, STEAP1, DLL3, bispecific antibody, cytokine release syndrome (CRS), molecular engineering, combination therapy

## Abstract

Advanced prostate cancer (PCa) is often resistant to standard immunotherapy due to its “immune-cold” environment. T-cell engagers (TCEs) offer a new frontier by directly linking T-cells to cancer cells via specific tumor antigens (like PSMA, STEAP1, and DLL3), bypassing traditional immune recognition. Early TCEs showed potential but faced toxicity and limited response durability. Newer agents, notably STEAP1-targeted TCEs, demonstrate more promising objective responses. Despite this progress, key challenges remain, including managing cytokine release syndrome (CRS), overcoming resistance mechanisms like antigen escape, and selecting the right patients. Future efforts will focus on improving TCE design, using rational combinations, identifying biomarkers, and integrating these therapies earlier in treatment. TCEs hold significant promise as a new pillar for advanced PCa, requiring continued innovation for optimal efficacy and safety.

## 1. Introduction

Prostate cancer (PCa) remains a leading cause of cancer-related deaths among men globally, with metastatic castration-resistant PCa (mCRPC) representing an incurable stage despite sequential therapies including androgen deprivation therapy (ADT), next-generation androgen receptor (AR) inhibitors, taxane chemotherapies, and radioligand therapy (RLT, e.g., 177Lu-PSMA) [1]. The transformative impact of immunotherapy seen in other malignancies has largely not extended to PCa [2]. This refractoriness is primarily attributed to its immunologically “cold” tumor microenvironment (TME), characterized by poor T-cell infiltration and low antigen presentation, rendering immune checkpoint inhibitors (ICIs) largely ineffective for most patients [3]. While offering an early glimpse into immunotherapeutic possibilities, the autologous vaccine sipuleucel-T—an immunotherapy where patient-derived antigen-presenting cells are activated ex vivo with a fusion protein targeting prostatic acid phosphatase (PAP)—showed only modest benefits, underscoring the need for more potent strategies [4].

T-cell engagers (TCEs) have emerged as a powerful alternative. Typically designed as bispecific antibodies, TCEs simultaneously bind T-cells (via CD3) and tumor-associated antigens (TAAs) on cancer cells. This forced synapse triggers direct, potent T-cell-mediated cytotoxicity, crucially bypassing the need for conventional T-cell receptor recognition and antigen presentation on major histocompatibility complex (MHC) molecules [5,6]. This mechanism is particularly advantageous in PCa, where MHC downregulation is a common immune evasion tactic, allowing TCEs to potentially activate potent killing even within poorly immunogenic tumors [7]. Initial efforts targeting prostate-specific membrane antigen (PSMA) provided essential proof-of-concept for TCEs in PCa [8] but also highlighted significant challenges regarding durability and toxicity. However, the field is rapidly advancing. Newer TCEs targeting other antigens like six transmembrane epithelial antigen of the prostate 1 (STEAP1) [9] and delta-like ligand 3 (DLL3) [10], often incorporating sophisticated engineering features, are showing increasingly promising clinical activity in refractory mCRPC.

This rapidly evolving landscape necessitates a timely synthesis and critical perspective. Therefore, this review aims to move beyond a mere compilation of trial data. We provide molecular insights into the diverse TCE strategies being employed against PCa (targeting PSMA, STEAP1, and DLL3), critically evaluate their associated clinical outcomes, and synthesize the lessons learned thus far. Our central argument is that while TCEs offer undeniable promise, transforming this potential into a robust and durable clinical reality for patients with advanced PCa requires strategically addressing inherent limitations through continuous innovation in molecular engineering and the implementation of evidence-based combination approaches.

Our narrative review begins with the fundamental mechanisms of TCE action and the innovative engineering approaches designed to harness their power while mitigating risks. We then progress through a critical examination of the evolving clinical landscape, analyzing trials targeting different antigens and evaluating their therapeutic value. Subsequently, we delve into the practical challenges of safety and toxicity management, compare TCEs against the existing standards of care, and culminate by outlining the crucial future directions—spanning novel targets, advanced constructs, biomarker development, and combination strategies—necessary to solidify the role of TCEs as a cornerstone of future PCa treatment and realize the potential argued herein.

## 2. Mechanism of Action and Novel Engineering Approaches of T-Cell Engagers

For clarity, ‘T-cell Engager (TCE)’ will refer broadly to constructs redirecting T-cells via CD3 and a tumor antigen. ‘BiTE^®^’ (Bispecific T-cell Engager), a registered trademark of Amgen Inc. (Thousand Oaks, CA, USA), denotes a specific, early TCE format comprising two linked scFvs, typically lacking an Fc region and having a short half-life. Thus, BiTEs^®^ are a subtype of TCEs.

### 2.1. Mechanism of Action

The fundamental mechanisms by which various T-cell-engaging cancer immunotherapies, including TCEs, ICIs, and chimeric antigen receptor (CAR)-T-cells, exert their effects are illustrated in Figure 1.

TCEs are engineered proteins, typically bispecific antibodies, designed to simultaneously bind T-cells and tumor cells. One arm recognizes CD3 on T-cells, while the other binds a tumor-associated antigen (TAA) expressed on PCa cells. By forming this physical bridge, TCEs induce an artificial immune synapse, triggering T-cell activation and directing cytotoxicity specifically towards the targeted tumor cells [11,12]. Crucially, this mechanism bypasses the requirement for peptide antigen presentation on MHC molecules for T-cell recognition [13]. Prostate tumors frequently downregulate MHC expression as an immune evasion tactic, thereby impairing conventional T-cell responses [7]. TCEs overcome this limitation by directly engaging T-cells for MHC-independent tumor cell killing, effectively redirecting T-cell activity based on target antigen presence.

Once engaged, activated T-cells release cytotoxic molecules like perforin and granzymes, along with pro-inflammatory cytokines (e.g., tumor necrosis factor alpha (TNFα), interleukin (IL)-6), at the tumor interface, leading to cancer cell lysis [6,8]. This process can potentially transform an immunologically ‘cold’ tumor microenvironment (TME) into an inflamed or ‘hot’ environment by promoting the influx and activation of T-cells and cytokines locally. This mode of action distinctly differs from that of immune checkpoint inhibitors (ICIs), which function by releasing inhibitory signals (e.g., programmed cell death protein 1 (PD-1)/programmed death-ligand 1 (PD-L1) blockade) on pre-existing T-cells [8]. Checkpoint blockade is effective only when tumor-specific T-cells are already present within the TME. In PCa, where effector T-cells are often scarce, releasing these “brakes” yields limited results for most patients [8,14]. In contrast, TCEs redirect T-cells already present in or trafficking through the tumor to engage cancer cells, initiating an immune attack irrespective of pre-existing T-cell specificity, thereby overcoming a key limitation of ICI therapy in this setting [15].

Compared to therapeutic cancer vaccines aiming to prime de novo immunity over time, TCEs provide immediate effector function by co-opting the patient’s existing T-cell repertoire. TCEs also differ from adoptive cell therapies using CAR T-cells. Adoptive CAR-T cell therapy involves extracting, genetically engineering, and reinfusing patient T-cells [16]. While sharing the goal of redirecting T-cells, TCEs are “off-the-shelf” pharmaceuticals, eliminating the need for personalized cell manufacturing and allowing for more accessible administration and dose adjustments. Furthermore, the effects of TCEs are pharmacologically reversible upon treatment cessation as the drug clears, whereas persistent CAR-T cells can lead to prolonged toxicity if severe adverse events occur. Early TCE formats (like the original BiTE^®^ (Bispecific T-cell Engager)) had a small size, leading to rapid clearance, necessitating continuous infusion [8]. Newer designs incorporate half-life extension strategies (e.g., fragment crystallizable (Fc) fragments, albumin-binding domains), permitting intermittent dosing (weekly or biweekly) while maintaining T-cell engagement and therapeutic effect [17]. This has significantly improved the clinical practicality of TCE therapy for solid tumors. In summary, this distinct mechanism of action—redirecting a patient’s own T-cells for immediate, targeted cytotoxicity—underpins the potential for TCEs to add a critical new dimension to the treatment paradigm for advanced PCa. However, this potent mechanism, relying on broad T-cell activation, also carries inherent challenges, primarily concerning safety and pharmacokinetics.

### 2.2. Novel Engineering Approaches

Addressing these intrinsic challenges became the primary driver for the sophisticated engineering approaches developed over the past decade. The evolution of these TCE formats, from simpler fragment-based constructs to more complex IgG-like and multispecific designs, is depicted in Figure 2. Early TCEs, such as those based on the bispecific T-cell engager (BiTE^®^) format, were typically small, Fc-free antibody fragments comprising two linked single-chain variable fragments (scFvs). While potent, their short circulating half-lives necessitated continuous infusion to maintain therapeutic levels [8]. To overcome this significant practical limitation, numerous engineering strategies have emerged over the past decade [18,19]. These strategies aim to enhance pharmacokinetic properties, mitigate immunogenicity, and reduce safety risks like cytokine release syndrome (CRS).

The figure illustrates the progression of TCE designs, many of which have been adopted by various T-cell engagers currently in clinical development, with examples such as Pasotuxizumab, Tarlatamab, Acapatamab, Xaluritamig, and HPN424 often representing these respective categories as indicated in the visual. These designs move from simpler to more complex formats with evolving therapeutic properties. The bispecific format includes fragment-based constructs (e.g., bispecific T-cell engager, BiTE^®^), which are typically composed of two single-chain variable fragments (scFvs), one targeting CD3 on T-cells and the other targeting a tumor-associated antigen on cancer cells; drugs like Pasotuxizumab and Tarlatamab exemplify this category. Key characteristics of these fragment-based formats include a small size enabling good tissue penetration, a short half-life often requiring continuous infusion, potent T-cell activation (with an associated risk of cytokine release syndrome—CRS), and an absence of Fc-mediated effector functions. Another bispecific format is the Fc-based construct (e.g., IgG heterodimer), such as Acapatamab or Xaluritamig, which incorporates an Fc region, leading to an extended half-life and less frequent dosing. These Fc-based formats offer the potential for Fc-mediated effector functions, such as antibody-dependent cell-mediated cytotoxicity (ADCC) and complement-dependent cytotoxicity (CDC), the strength of which varies by the chosen IgG subclass (IgG1, IgG2, IgG3, IgG4). The choice of IgG subclass is critical as it tunes these effector functions (e.g., higher ADCC/CDC with IgG1/IgG3 versus lower with IgG2/IgG4) and pharmacokinetic profiles. However, this format results in an increased size, which may impact tissue penetration, and slightly increased manufacturing complexity. The multispecific format, with HPN424 being an illustrative example of a trispecific construct, represents more advanced designs targeting multiple antigens or incorporating additional functionalities. Trispecific killer engagers, multispecific antibodies, and bispecific antibody drug conjugates (bispecific ADCs) aim for enhanced specificity and/or efficacy through multiple antigen targeting. They may also provide T-cell co-stimulation (potentially increasing efficacy and reducing T-cell exhaustion) and offer an improved safety profile through mechanisms like conditional activation or more precise tumor-specific targeting. However, these advanced formats come with significantly increased design and manufacturing complexity. The overall evolution reflects efforts to improve efficacy, extend half-life, modulate effector functions, enhance safety, and overcome resistance mechanisms in cancer immunotherapy. Created with BioRender.com. Abbreviations: ADCC, antibody-dependent cell-mediated cytotoxicity; ADC, antibody-drug conjugate; BiTE, bispecific T-cell engager; CD, cluster of differentiation; CDC, complement-dependent cytotoxicity; CRS, cytokine release syndrome; Fc, fragment, crystallizable; IgG, immunoglobulin G; scFv, single-chain variable fragment; TCE, T-cell engager.

Recent developments have focused primarily on two format types: immunoglobulin G (IgG)-like and fragment-based constructs. IgG-like formats incorporate engineered Fc domains, often featuring mutations to abrogate Fc receptor binding (silencing effector functions) while leveraging the FcRn interaction to extend circulating half-life, enabling intermittent dosing schedules [17]. Prominent examples include “2 + 1” or “CrossMab” formats with two tumor antigen-binding arms and one CD3-targeting domain, employing technologies like “knob-into-hole”—where steric clashes are engineered by introducing a ‘knob’ (a large amino acid) into one heavy chain and a ‘hole’ (a smaller amino acid) into the other, favoring heterodimerization over homodimerization— or other chain-pairing strategies to ensure correct assembly and enhance tumor selectivity [20]. Alternatively, fragment-based constructs lacking an Fc domain may offer superior tumor penetration due to their smaller size but often integrate albumin-binding domains or other moieties to prolong serum persistence [21].

Affinity tuning represents another critical engineering advance [22]. Modulating the binding affinities for CD3 and the tumor antigen allows for precise control over T-cell activation, potentially confining potent activation primarily to the TME and thereby reducing systemic CRS risk. Furthermore, concepts like “masked” or prodrug-like TCEs, which remain systemically inert and are activated selectively within the tumor microenvironment (e.g., by tumor-associated proteases), are emerging as a means to significantly improve the safety profile [18,23]. Advanced multispecific designs are also under investigation. These include trispecific constructs targeting multiple tumor antigens (to overcome heterogeneity or escape) or integrating costimulatory signals (e.g., targeting 4-1BB (CD137)) or cytokine moieties (e.g., IL-15) to provide synergistic effects by simultaneously addressing antigen escape and boosting immune activation or survival [24,25]. Collectively, these sophisticated engineering innovations are substantially advancing the clinical applicability and potential of TCEs, particularly for challenging solid tumors like PCa, by optimizing therapeutic potency, safety, and response durability.

## 3. Clinical Development and Therapeutic Value of TCEs for Prostate Cancer

Multiple TCEs targeting PCa antigens have entered clinical trials with formats ranging from traditional BiTE molecules to trispecific T-cell engagers (TriTEs) designed for greater stability and half-life, for example, by incorporating Fc domains for improved structural integrity and FcRn-mediated recycling or by utilizing other sophisticated protein engineering techniques that enhance the construct’s overall robustness compared to simpler, Fc-free formats. Table 1 summarizes the main TCEs currently in clinical development targeting PSMA, STEAP1, or DLL3 in advanced PCa. Early trials demonstrated that TCEs can induce prostate-specific antigen (PSA) decline and objective response rate (ORR) in patients with mCRPC who have exhausted conventional therapies, albeit with some challenges in tolerability and durability [12,26,27].

### 3.1. Key Insights from Approved TCEs in Hematology

TCEs approved for hematologic malignancies have rapidly reshaped the treatment landscape, offering insights that can inform PCa immunotherapy. First, robust efficacy can be achieved in late-line heavily pretreated populations, underscoring the potency of redirecting a patient’s own T-cells. Trials of agents such as blinatumomab [31], glofitamab [32], and teclistamab [33] have demonstrated that response rates can exceed those of traditional chemotherapy in refractory settings, even producing durable remission in some patients. Second, the safety profiles of these agents highlight the critical role of cytokine release syndrome (CRS) monitoring and early intervention [7,17,20,34]. Although step-up dosing and tocilizumab prophylaxis effectively temper high-grade CRS, these protocols must be carefully tailored to the pharmacological features and disease context of each TCE. Third, antigen escape and downregulation remain frequent mechanisms of acquired resistance [6,8,32,35]; combining TCEs with other treatments (e.g., checkpoint inhibitors or alternative bispecific constructs) can potentially preserve long-term disease control [36]. Finally, many TCEs have received accelerated or conditional approval based on clinically meaningful response rates in niche-refractory populations, demonstrating the regulatory path that new PCa-directed TCEs may follow. Although prostate tumors present a distinct immune environment relative to B-cell malignancies, hematological experience with approved TCEs provides a valuable blueprint for optimizing dosing regimens, toxicity management, and trial designs in solid tumor settings.

### 3.2. PSMA-Targeting TCEs

PSMA (prostate-specific membrane antigen, also known as FOLH1), a type II transmembrane glycoprotein with carboxypeptidase enzymatic activity, became the initial primary target for TCE development in this disease. While highly expressed on most PCa cells—with expression generally increasing with tumor grade, metastatic progression, and after ADT—its physiological roles include nutrient uptake and putative involvement in cell migration and neovascularization. Limited normal tissue distribution (e.g., salivary glands, kidney, small intestine) contributes to its therapeutic appeal. Importantly, PSMA frequently co-expresses with other PCa antigens like STEAP1, a factor relevant for potential combination or sequential targeting strategies [37]. Early proof-of-concept was established by pasotuxizumab (AMG 212), a classic BiTE molecule [8]. While demonstrating the potential for TCEs to induce >50% PSA declines in some mCRPC patients, its clinical utility was significantly hampered by a short half-life necessitating continuous infusion, notable immunogenicity, and importantly, a lack of durable responses [8,12,17,38]. These initial learnings directly spurred the development of second-generation PSMA TCEs engineered for extended half-lives and potentially improved tolerability.

Acapatamab (AMG 160), incorporating an IgG-like Fc domain, exemplified this next wave [12]. In its Phase I trial, it showed promising pharmacokinetic profiles and achieved >50% PSA reduction in 63% of evaluable patients treated at higher doses [12,39]. However, this biochemical activity did not translate into high rates of objective radiographic response, highlighting a potential disconnect often observed in mCRPC trials and underscoring the need for further optimization, potentially in molecular design (e.g., CD3 affinity tuning) or patient selection beyond just PSMA presence. Cytokine release syndrome remained frequent, though reported as mostly manageable with mitigation strategies [39].

Other approaches followed, such as HPN424, a trispecific TCE targeting PSMA, CD3, and albumin for further half-life extension [39]. Early Phase I data indicated modest activity, with only about 20% of patients showing any PSA decline and ~5–6% achieving a ≥50% reduction, although stable disease (SD) was noted in roughly half. While CRS was again prevalent, step-up dosing rendered it largely low-grade, reinforcing the importance of administration schedules [39]. Collectively, these early-generation PSMA TCE trials, while validating the target and mechanism, revealed persistent hurdles in achieving deep and durable objective responses and consistently managing CRS without compromising efficacy. This context motivated exploration of novel constructs such as the fully human CC-1. One such promising agent is CC-1, a fully human bispecific antibody targeting CD3 and PSMA (Table 1), whose fully human composition represents a significant engineering advancement. Unlike murine or chimeric antibodies, which can elicit anti-drug antibody (ADA) responses in patients, a fully human antibody is expected to have lower immunogenicity. This reduced immunogenic potential can translate to improved safety, better pharmacokinetic profiles due to less rapid clearance, and sustained efficacy. CC-1 showed encouraging early signals of universal PSA decline (up to 60%) with predominantly mild CRS in a small Phase I cohort [28]. This context also motivated targeting alternative immune effector cells. For instance, LAVA-1207 is a bispecific TCE that, instead of a pan-T-cell CD3-binding arm, employs one arm to directly bind the Vgamma9 Vdelta2 T-cell receptor (Vγ9Vδ2 TCR), thereby selectively engaging this particular subset of gamma delta T-cells for PSMA-targeted tumor cell lysis [30]. More crucially, these challenges also intensified interest in alternative TAAs, such as STEAP1. While PSMA targeting provided critical initial validation, the quest for improved efficacy and strategies to overcome potential resistance mechanisms spurred the investigation of other promising antigens like STEAP1.

### 3.3. STEAP1-Targeting TCEs

STEAP1 (six transmembrane epithelial antigen of the prostate 1) emerged as another compelling TAA. It is a cell-surface transmembrane protein whose functions are thought to involve intercellular communication and potentially modulation of cell proliferation and apoptosis. Given its overexpression in approximately 80–95% of metastatic PCa and minimal normal tissue expression, it presents a broad therapeutic window. Dynamic changes in STEAP1 expression during PCa progression or treatment are still being fully characterized. Notably, STEAP1 is often co-expressed with PSMA on prostate cancer cells, suggesting its utility in patients with heterogeneous PSMA expression or as part of multi-antigen targeting approaches [9,40]. The STEAP1-directed TCE AMG 509 (Xaluritamig) generated considerable excitement with its first-in-human trial results. In a Phase I study encompassing 97 heavily pretreated mCRPC patients, AMG 509 achieved a PSA50 response in 49% overall, rising to 59% in higher-dose cohorts, with 36% achieving ≥90% PSA reductions. Importantly, objective tumor responses per Response Evaluation Criteria In Solid Tumors (RECIST) criteria were observed in 24% (41% at high doses), including deep regressions in visceral sites [19].

These efficacy figures, particularly the ORR, appear notably higher than those reported in many initial Phase I trials of PSMA-targeted TCEs, which often struggled to exceed 20% ORR [12]. While direct cross-trial comparisons are inherently limited by patient heterogeneity and trial design, this apparent disparity warrants consideration. Potential contributing factors could include (1) STEAP1’s potentially broader and more homogeneous expression across mCRPC lesions compared to PSMA in some populations, providing more consistent target availability [41]; (2) superior molecular engineering or optimized binding affinities inherent to the Xaluritamig construct itself [19]; or (3) differences in target biology. For instance, relatively slower internalization kinetics of STEAP1, compared to other potential targets, could lead to its prolonged availability on the cell surface. This, in turn, might enhance TCE engagement, promote more sustained immune synapse formation, and thereby contribute to the observed T-cell-mediated cytotoxicity with xaluritamig. Alternatively, distinct downstream signaling pathways initiated upon STEAP1 engagement by the TCE might also play a role. Further investigation, including potential future head-to-head studies, is needed to dissect these possibilities.

The safety profile, while showing class-typical CRS in ~72% of patients, was deemed manageable [19]. Critically, the low incidence of grade (G) ≥3 CRS (two cases) and treatment discontinuation due to CRS (3%) suggests that the implemented mitigation strategies (e.g., priming and step-up dosing) were particularly effective for this agent in this trial. However, for a comprehensive view of treatment perseverance, it is important to note that discontinuations due to other treatment-emergent adverse events (TEAEs), aside from CRS, were also observed, with myalgia accounting for approximately 3% of discontinuations and arthralgia for approximately 2% [42]. This broader context, while still potentially offering a more favorable overall tolerability signal compared to some earlier TCE experiences, is critical for assessing the agent’s complete safety and manageability.

Although anti-drug antibodies (ADA) were detected, their lack of correlation with efficacy suggests responses occurred rapidly, prior to significant ADA interference [19]. These robust early data, representing a potential step-change in TCE efficacy for PCa, have justifiably propelled AMG 509 towards pivotal Phase III evaluation [42].

Beyond the common adenocarcinoma histology, a subset of advanced PCa exhibits neuroendocrine features, presenting distinct therapeutic challenges and necessitating novel targets such as DLL3.

### 3.4. Delta-like Ligand 3 (DLL3)-Targeting TCEs in Neuroendocrine Prostate Cancer

A subset of advanced PCa can undergo neuroendocrine differentiation, either de novo or as a result of the evolution of treatment-resistant mCRPC [16]. Neuroendocrine prostate cancer (NEPC) typically loses androgen receptor expression and PSA production, behaves aggressively, and has a poor prognosis [16]. DLL3 is a cell surface protein expressed in neuroendocrine tumors (notably small cell lung cancer (SCLC) and NEPC) but not in normal adult tissues, making it an ideal immunotherapeutic target in this context. DLL3 is an atypical Notch pathway ligand that, in contrast to canonical Notch ligands, is predominantly localized to the Golgi in normal cells but exhibits aberrant cell surface expression in neuroendocrine tumor cells, where its functional role on the surface is an area of active research. Its expression in NEPC often occurs as these tumors evolve from adenocarcinoma and lose expression of typical markers like AR and PSA. Consequently, DLL3 is usually not co-expressed with PSMA or STEAP1, marking it as a target for a distinct, aggressive PCa subtype. The consistency and level of DLL3 expression and its modulation over time or with therapy in NEPC are critical factors for the success of DLL3-targeted agents [16,43]. Tarlatamab (AMG 757) is a half-life extended (HLE) BiTE^®^ that targets DLL3 in tumor cells and CD3 in T-cells. Unlike the original short-acting BiTE^®^ format, HLE BiTEs^®^ like tarlatamab incorporate molecular engineering strategies, such as the addition of an Fc domain or other moieties, to prolong their circulating half-life, thereby permitting less frequent (e.g., intermittent) dosing schedules while maintaining therapeutic engagement. It was first developed for small-cell lung cancer and has shown anti-tumor activity [10]; it has recently been tested in NEPC [16]. In a Phase I trial presented at the American Society of Clinical Oncology (ASCO) 2024, patients with NEPC (either treatment-emergent or de novo NEPC with genomic loss of TP53, RB1, or PTEN) received tarlatamab at doses up to 100 mg intravenously (IV) every 2 weeks (Q2weeks). The efficacy in this PCa cohort was modest but notable given the refractory setting [16]. The ORR was 10.5% and 22.2% among the subsets of patients with DLL3-positive tumors. DLL3-positivity was defined as ≥1% expression by immunohistochemistry (IHC); about 56% of patients met this low threshold). Most of the responses were partial and not particularly durable. The median treatment duration was only 1.4 months in the overall cohort, indicating that many patients progressed early, whereas responders in the DLL3-high group had a median response duration of ~7.3 months. However, one patient had an exceptionally long-lasting response lasting beyond 2 years, illustrating that profound and durable remission is possible in principle. Investigators suggest that better patient selection may improve outcomes, as treating patients with low or negative DLL3 expression likely diluted the overall efficacy results. In terms of safety, the NEPC tarlatamab study recorded CRS in 75% of patients (mostly grade 1–2, only a single grade ≥3 case) and a low incidence of neurotoxicity (12.5%, with one grade 3 event). These toxicities are reminiscent of experiences in small cell lung cancer and appear manageable with appropriate precautions [16]. While the activity of DLL3 TCEs in NEPC has yet to reach the high response rates seen with other targeted agents in different settings, this trial provided proof of concept that T-cell redirection can target an aggressive variant of PCa that is otherwise extremely difficult to treat. Ongoing efforts will likely refine this approach (e.g., by requiring higher DLL3 expression or combining it with other therapies) to increase its impact. Given the lack of options for NEPC, identifying a subset of patients benefiting from it (as seen here) is a meaningful step forward.

### 3.5. Emerging Efficacy Landscape and Intrinsic Therapeutic Value of TCEs in mCRPC

Synthesizing the early clinical data presented for TCEs targeting PSMA, STEAP1, and DLL3 (Table 1), a compelling picture emerges: T-cell engagers possess the intrinsic capability to elicit meaningful anti-tumor activity in heavily pretreated mCRPC patients, a population often refractory to standard therapies. Biochemical responses (PSA declines ≥50%) have been consistently observed across various constructs, ranging roughly from 20% to over 60% in specific cohorts [8,12,19]. Crucially, objective tumor responses (ORR according to RECIST) have also been documented, with rates spanning from approximately 10% in some initial PSMA trials to encouraging figures exceeding 40% in higher-dose cohorts of the STEAP1-targeting Xaluritamig study [12,19].

While these early results must be interpreted with caution—stemming largely from non-randomized Phase I trials with inherent patient heterogeneity and often limited follow-up—the ability to achieve objective regressions, including in visceral sites, via an off-the-shelf immunotherapy represents a significant potential advance in the ‘immune-cold’ landscape of PCa. This activity, driven by direct T-cell redirection independent of MHC presentation [5], offers a unique mechanistic advantage where conventional immunotherapies like ICIs have largely failed outside niche populations [44]. The higher response rates suggested by the STEAP1-targeted approach compared to some initial PSMA TCEs (though direct comparisons are premature) hint at the critical influence of target selection [15] and potentially optimized molecular engineering [45] on clinical outcomes.

However, realizing the full therapeutic value of TCEs necessitates confronting key challenges highlighted by these initial studies. Foremost among these is the question of response durability [6]. While some patients experience prolonged disease control, early relapse or primary resistance remains common, likely driven by factors such as antigen loss or T-cell exhaustion that require further investigation and specific counter-strategies [6,46]. Furthermore, optimal patient selection remains a critical unmet need. While target antigen expression (via PET or IHC) is a prerequisite, the ideal expression threshold, the predictive value of antigen density, and the influence of the tumor microenvironment (e.g., baseline T-cell infiltration, immunosuppressive factors) on TCE efficacy are still poorly understood [47]. The modest activity of tarlatamab in a broadly defined DLL3-positive NEPC population [48] underscores that simply identifying the target antigen may be insufficient for predicting meaningful benefit.

In essence, the current evidence provides strong proof-of-concept for the therapeutic value of TCEs in mCRPC, demonstrating their ability to induce responses in refractory disease. Yet, variability in efficacy across agents and targets, unanswered questions about durability, and the need for refined predictive biomarkers clearly indicate that significant work remains to translate this potential into consistent, long-term clinical benefit. Understanding how this unique therapeutic modality compares and integrates with established treatments is the next critical step.

## 4. Safety and Toxicity Mitigation Strategies

The promising anti-tumor activity highlighted in the previous section can only be translated into true clinical benefit if the associated toxicities, inherent to potent T-cell activation, are effectively managed.

Cytokine release syndrome (CRS) and neurotoxicity (Immune Effector Cell-Associated Neurotoxicity Syndrome (ICANS)) are major challenges [49]. Although TCEs offer a powerful therapeutic strategy, their administration is often associated with significant adverse events, including CRS, ICANS, and immunogenic complications, requiring robust safety measures. CRS results from extensive T-cell activation and the subsequent release of cytokines, such as IL-6, TNFα, and IL-1, leading to clinical manifestations including fever, hypotension, hypoxia, and organ dysfunction [49,50]. The risk of CRS correlates with tumor burden, dosage, and CD3 binding affinity of the TCE construct; thus, patients with widespread metastatic PCa lesions are particularly susceptible at therapy initiation. To address this, step-up dosing regimens involving initial low-dose priming followed by incremental increases are used to effectively reduce cytokine surges. Pharmacological interventions, such as tocilizumab, corticosteroids, and IL-1 inhibitors like anakinra, are frequently employed at early CRS onset, significantly mitigating progression to severe presentations [49,51]. ICANS poses an additional risk, characterized by symptoms ranging from mild encephalopathy and confusion to severe seizures and focal neurological deficits. Vigilant neurological monitoring and prompt initiation of high-dose corticosteroids are critical for managing moderate-to-severe cases [17], and temporary suspension or discontinuation of TCE therapy may be necessary if symptoms persist or worsen. Historically, ADA has limited the efficacy of early-generation TCEs, often due to insufficient humanization, resulting in shortened half-lives and reduced therapeutic potency [8]. Modern TCE designs leverage human or humanized frameworks with engineered variable regions, thereby significantly reducing the number of immunogenic epitopes. Additionally, intermittent dosing regimens, rather than continuous infusion, further reduce immunogenic responses [52]. Off-tumor toxicity is addressed primarily by selecting prostate-specific antigens, such as PSMA and STEAP1, to minimize risks to non-prostate tissues [40]. However, low antigen expression in normal tissues requires careful monitoring. Advanced mitigation strategies include the development of masked or prodrug TCE constructs, which are engineered TCEs rendered temporarily inactive or with blocked binding sites. These are designed to circulate systemically in an inert form and become selectively activated preferentially within the tumor microenvironment due to local conditions like specific enzymatic activity (e.g., from tumor-associated proteases such as MMPs or FAP), acidic pH, or hypoxia, which cleave masking elements or convert the prodrug. This targeted activation confines potent TCE activity primarily to the tumor site, thereby further lowering the risk of off-target effects and improving the therapeutic index [18,53]. Combination strategies involving TCEs may increase efficacy but carry an increased risk of CRS and immune-mediated adverse events [17]. Protocols often incorporate sequential approaches, initially employing cytoreductive treatments such as RLT to reduce tumor burden before introducing TCEs, thus minimizing the magnitude of cytokine release. Close monitoring through laboratory evaluations, including cytokine profiles, and complete blood counts, and coordinated multidisciplinary care involving oncology, hospital medicine, and critical care units ensures early detection and effective management of immune-related toxicities. With the continued clinical integration of TCE therapy in advanced PCa, refined step-up dosing regimens, proactive pharmacological interventions, improved immunogenicity profiles, and precision-targeting technologies promise a progressively safer therapeutic landscape.

With optimized dosing regimens and proactive management protocols increasingly demonstrating the feasibility of mitigating major toxicities like CRS, the question then becomes how TCEs, with their distinct efficacy and safety profiles, compare and potentially integrate with the existing therapeutic landscape for advanced PCa.

## 5. Comparison with Existing Therapies

Having established the potential efficacy, intrinsic value, and key challenges of TCEs based on early clinical evidence, it is crucial to position this novel therapeutic class within the existing treatment landscape for advanced PCa. TCEs offer distinct mechanisms, efficacy profiles, and toxicity considerations compared to established modalities, as summarized in Table 2. Understanding these differences is essential for determining their optimal role and integration into future treatment algorithms.

### 5.1. Comparison with Androgen Receptor (AR)-Targeted Agents

TCEs provide an AR-independent mechanism, crucial for patients progressing on agents like enzalutamide or abiraterone [54]. These ARTAs, when used in earlier lines for mCRPC, yield PSA50 response rates typically ranging from ~20–40% and radiographic progression-free survival of several months [58,59,60]. However, resistance invariably develops. Potential synergy exists, as AR modulation may alter PSMA expression [55], but TCEs fundamentally bypass AR resistance pathways. Toxicity profiles differ significantly (immune-mediated for TCEs vs. metabolic/fatigue for AR agents) [61].

### 5.2. Comparison with Chemotherapy (Taxanes)

Chemotherapy, such as docetaxel or cabazitaxel, offers broad cytotoxicity but carries risks of significant systemic toxicities (e.g., myelosuppression, neuropathy, alopecia). In heavily pretreated mCRPC patients (often third-line or later, a population where TCEs are frequently investigated), cabazitaxel has shown objective response rates (ORR) in the range of 10–20% and a median overall survival benefit of a few months [19,62,63]. In contrast, TCEs induce targeted T-cell killing but have distinct risks like CRS and ICANS. While direct cross-trial comparisons are challenging, some early TCE data (particularly STEAP1-targeted) show ORR in late-line settings that appear competitive with, or potentially exceed, these historical chemotherapy benchmarks. Chemotherapy benefits are often transient; TCE durability remains a key area of investigation. Pre-TCE cytoreduction with chemo to potentially lower CRS risk is an area of interest [64].

### 5.3. Comparison with Radioligand Therapy (177Lu-PSMA-617)

Both require PSMA expression (for PSMA TCEs/RLT). 177Lu-PSMA-617 (RLT) delivers targeted radiation, has proven survival benefits in Phase III (e.g., the VISION trial in PSMA-positive mCRPC patients progressing after ARTAs and taxanes reported ~30% ORR and ~46% PSA50 response rates), and has distinct toxicities (hematologic, xerostomia) [56]. Early TCE trials (e.g., Xaluritamig targeting STEAP1, potentially relevant for PSMA-low patients [9,19]) suggest ORRs that might rival those seen in VISION [56], although TCE data are less mature. Furthermore, potential resistance mechanisms differ (radiation vs. immune escape) [52,65]. Sequential or combination approaches are highly anticipated but require careful study regarding feasibility and additive toxicity.

### 5.4. Comparison with Immune Checkpoint Inhibitors (ICIs)

Unlike ICIs, which are largely ineffective in unselected mCRPC (typically showing ORRs <5–10% except in biomarker-selected populations like MSI-H/dMMR patients) due to the ‘cold’ TME [66], TCEs actively redirect T-cells, bypassing the need for pre-existing anti-tumor immunity required for ICI efficacy [11]. Their mechanisms are thus complementary, raising significant interest in combination strategies to potentially sustain TCE-activated T-cell function [11]. Toxicity profiles are both immune-related but manifest differently (e.g., CRS is unique to TCEs/CAR-T) [17,67].

### 5.5. Comparison with Sipuleucel-T

Relative to Sipuleucel-T, which aims to prime an immune response over time and demonstrated a modest median overall survival benefit of ~4 months with minimal PSA impact or objective responses in asymptomatic or minimally symptomatic mCRPC patients [4,57], TCEs induce immediate, direct cytotoxicity via pre-existing T-cells [68]. They represent fundamentally different immunotherapeutic approaches in terms of mechanism, potency, and kinetics.

In summary, TCEs occupy a unique position. They offer potent, targeted immunotherapy for a historically ‘immune-cold’ tumor, acting independently of AR signaling and conventional MHC presentation. Their efficacy signals in heavily pretreated patients appear promising relative to some existing options, particularly chemotherapy in later lines, although data maturity and durability remain key considerations. Crucially, their distinct mechanism and non-cross-resistance profile suggest significant potential for integration—either sequentially or concurrently—with AR therapies, chemotherapy, or RLT. However, their unique immune-mediated toxicities necessitate specialized management. Defining the optimal place for TCEs—whether as salvage therapy, sequenced with RLT, combined with ICIs, or even moved into earlier disease settings—will depend on results from ongoing and future randomized controlled trials that directly compare these strategies. Nevertheless, their demonstrated ability to generate objective responses where other immunotherapies have faltered strongly positions TCEs as a likely future pillar in the management of advanced PCa. However, this comparison also underscores that TCEs are unlikely to be a universal solution alone. Addressing challenges like durability and reaching broader patient populations necessitates exploring how to best integrate their unique strengths with other modalities, paving the way for the future research directions discussed next.

## 6. Future Directions and Research Priorities

Positioning TCEs within the treatment landscape underscores their significant potential alongside the key hurdles that must be overcome as they advance toward broader clinical use in PCa. An overview of these future directions and research priorities essential for advancing TCE therapy in PCa is presented in Figure 3. To realize the full promise of this therapeutic class, several key research priorities have emerged, focusing on critical challenges identified thus far, including enhancing durability, improving safety profiles, overcoming resistance, and optimizing patient selection. Below, we outline these important directions for future research aimed at achieving TCE therapy’s full potential.

### 6.1. Broadening Target Antigens and TCE Platforms

While PSMA and STEAP1 are currently the principal targets, researchers are exploring additional TAAs to reach patients whose tumors have heterogeneous antigen expression. For instance, the prostate stem cell antigen was targeted by a TCE (GEM3PSCA) in early trials (NCT03927573). Human Kallikrein-related peptidase 2 (hK2), an antigen related to PSA, is another target, and Janssen’s JNJ-78278343 (a Kallikrein-related peptidase 2 (KLK2) xCD3 TCE) is in Phase I testing [29,69]. Targeting multiple antigens may be crucial for covering the full spectrum of PCa biology and preventing escape. In the future, multispecific engagers may simultaneously target two tumor antigens and CD3, thereby attacking heterogeneous tumor cell populations in one stroke [45]. Preclinical designs are being developed, such as dual-affinity retargeting bispecifics (constructs where binding affinities for two distinct targets, like a tumor antigen and CD3, are differentially optimized to fine-tune immune cell activation or enhance tumor specificity over normal tissues) and tandem trimeric engagers (TriKEs—multispecific molecules, often with three domains in a single chain, designed, for instance, to engage an immune cell activating receptor like CD16a on NK cells while binding two different tumor antigens, or to bridge a T-cell to a tumor antigen via CD3 while also delivering a co-stimulatory signal) [70]. Moreover, strategies are being developed that engage alternative immune effector cells, such as natural killer (NK) cells (further discussed in Section 6.2). These novel platforms aim to complement TCEs or provide alternatives when T-cell function is impaired. The pipeline of next-generation engagers is rich, and ongoing research will identify antigens and formats that yield the best therapeutic index.

Beyond refining existing formats, an innovative future direction for TCE platforms involves their in vivo production by genetically engineered patient cells. This strategy aims to have cells, such as T-cells, B-cells, or even hepatocytes, act as ‘biofactories’, continuously secreting TCEs directly within the body. Recent advancements in gene delivery technologies, particularly adeno-associated virus (AAV) vectors for potentially long-term expression and lipid nanoparticles (LNPs) for transient mRNA delivery or gene editing approaches, are bringing this concept closer to clinical reality [71].

This in vivo TCE production model offers several compelling perspectives [72,73]. It could ensure sustained and stable therapeutic levels of the engager, potentially overcoming the pharmacokinetic challenges and frequent dosing regimens associated with some exogenously administered TCEs. Furthermore, if immune cells trafficking to the tumor (like engineered T-cells) are utilized for secretion, localized TCE production within the tumor microenvironment could enhance anti-tumor activity while minimizing systemic side effects. However, this approach is not without significant challenges. Precise control over the rate and duration of in vivo TCE expression is paramount to mitigate risks of uncontrolled immune activation or ‘on-target, off-tumor’ toxicities. The safety of the gene delivery systems, potential immunogenicity against the engineered cells or the secreted TCE, and the long-term biological consequences of sustained cellular engineering require thorough investigation. Addressing these complexities, alongside navigating distinct manufacturing and regulatory hurdles, will be critical to realizing the therapeutic potential of these next-generation TCE delivery platforms.

### 6.2. Beyond CD3—NK Engagers and Multispecific Constructs

In addition to engaging CD3-positive T-cells, some newer bispecifics harness other immune cell subsets or combine multiple functionalities to augment their antitumor activity. One emerging approach involves bridging cancer cells to NK cells, typically by targeting CD16 (FcγRIIIa) on NK cells and a TAA on the tumor cell [74]. While TCEs also bypass the need for MHC presentation for their cytotoxic effect, NK cell engagers offer several distinct potential advantages. NK cells employ a range of cytotoxic mechanisms, including perforin/granzyme pathways and engagement of death receptors on tumor cells, which may be effective against tumor cells that have developed resistance to T-cell-mediated apoptosis. Furthermore, in patients where T-cell function is compromised or exhausted (e.g., after multiple lines of therapy or due to the tumor microenvironment), engaging NK cells could provide an alternative cytotoxic immune response. The spectrum and severity of toxicities, such as cytokine release syndrome (CRS), might also differ between NK cell engagers and TCEs due to the distinct cytokine profiles released by activated NK cells compared to T-cells. Additionally, NK cell engagers could potentially be used in combination with or sequentially to TCEs to target tumors through multiple immune effector pathways, possibly overcoming heterogeneous resistance mechanisms. These attributes make NK cell engagers a compelling strategy, particularly for tumors with downregulated HLA expression or defects in antigen processing where conventional T-cell responses (other than TCE-mediated) are impaired or as a complementary approach to T-cell redirection [74,75]. Early clinical experience with NK-engaging bispecifics (sometimes referred to as BKEs) has demonstrated manageable safety profiles and encouraging efficacy in hematological cancers, primarily by capitalizing on antibody-dependent cellular cytotoxicity (ADCC) [70]. Although still in the early stages, NK engagers can be applied to solid tumors, including PCa, particularly those that exhibit a “cold” TME or limited T-cell infiltration. Meanwhile, multispecific constructs expand beyond the two-arm model. Trispecific designs can incorporate a second TAA or costimulatory/cytokine arm. For instance, some formats couple CD3 and two different TAAs (“OR-gated” targeting) to reduce the risk of antigen-negative escape [76]. Others incorporate elements that provide a costimulatory signal—a secondary signal crucial for optimal T-cell (or NK cell) activation, proliferation, and survival, beyond the primary engagement of the activating receptor (like CD3 or CD16)—such as 4-1BB agonism, or add cytokine moieties like IL-15. These additions aim to enhance T-cell or NK cell expansion, persistence, and potency within the TME, thereby potentially leading to a more robust and durable anti-tumor response [76,77]. Trispecific constructs targeting PSMA, CD3, and an albumin-binding domain have already entered clinical trials for PCa (e.g., HPN424) [12], illustrating how multispecificity can simultaneously boost half-life and tumor specificity. Over time, these advanced formats may be refined to address the known pitfalls of single-target TCEs, such as partial antigen loss and suboptimal durability. Ultimately, by going beyond classical CD3-mediated redirection and incorporating additional targets or immune effector mechanisms, next-generation agents seek to improve both efficacy and safety, offering a broader immunological arsenal against advanced PCa.

### 6.3. OR-Gated vs. AND-Gated Targeting Strategies

OR-gated and AND-gated targeting strategies offer distinct ways to enhance the selectivity and efficacy of bispecific and multispecific therapies [78]. OR-gated designs target two different antigens and kill tumor cells expressing either antigen, thereby reducing the likelihood of immune escape through antigen loss or downregulation. This approach is especially relevant in advanced PCa, where the heterogeneous expression of targets such as PSMA or STEAP1 can hamper single-antigen treatments over time. In contrast, AND-gated approaches require the simultaneous binding of two distinct antigens on the same cell surface before triggering cytotoxicity. By necessitating the co-expression of both markers, AND-gating aims to limit on-target off-tumor toxicity, thereby improving safety, particularly in healthy tissues that might express one of the target antigens at low levels [78,79]. These designs, sometimes referred to as “split-antibody” or “hemi-body” concepts (where separate components of the therapeutic molecule, each targeting a different antigen, must assemble on the cancer cell surface to become fully active), are under active investigation as a means to exploit unique tumor-antigen combinations while sparing healthy tissues. In advanced PCa, dual-marker gating might be adapted to include a key TCE domain that only becomes active once both prostate-specific antigens are recognized, thus potentially achieving more precise killing of malignant cells while minimizing risks to the normal prostate or other nontarget tissues. Whether employing OR gating to broaden the coverage of heterogeneous tumors or AND gating to enhance specificity, these strategies exemplify next-generation innovations aimed at overcoming the core challenges faced by single-target TCEs.

### 6.4. Biomarker-Driven Patient Selection

Optimizing patient selection is fundamental to improving TCE outcomes. Expression of the target antigen seems to be the most obvious selection criterion. STEAP1 is expressed in proportionally more patients than PSMA in some analyses of mCRPC (~95% vs. ~68% in one analysis [15], or 87.7% vs. 60.5% [80]), suggesting that STEAP1 TCEs could treat a wider population. In trials, IHC or PET imaging is used to confirm the presence of the target (for example, PSMA PET scans ensure that a patient’s metastases express PSMA before receiving PSMA TCEs, analogous to selection for PSMA RLT) [47]. In the future, a combination of biomarkers may be used: antigen density (quantified by imaging uptake or histology) could predict the intensity of the TCE response, while immune contexture markers (such as baseline T-cell infiltration or interferon-gamma gene signatures) might predict how readily the TCE-redirected T-cells can infiltrate and function in a given tumor [26]. For DLL3-targeted therapy in NEPC, future trials will likely require a higher DLL3 expression cutoff to enrich for responders [48]. Another biomarker angle is the monitoring of immune activation markers during therapy (e.g., cytokine levels or circulating T-cell profiles) to gauge whether the patient is mounting a response and to preemptively manage side effects. As our understanding grows, we may develop a panel of biomarkers, including tumor genomics, surface antigen profiling, and immune phenotypes, to personalize TCE therapy, identify those most likely to benefit, and guide the choice of target antigen for each patient.

### 6.5. Enhancing Safety and Mitigating Toxicity

Alongside enhancing efficacy through novel platforms and better patient selection (discussed in Section 6.4), improving the safety profile remains paramount. Safety management remains the top priority in the development of TCEs. CRS, a primary safety concern, results from the widespread T-cell activation and massive cytokine release triggered by TCE engagement (as detailed in Section 4) and, while often manageable, is closely linked to the mechanism of action of TCEs and thus cannot be entirely eliminated. However, structural modifications are being implemented to reduce the severity of CRS without sacrificing efficacy. One strategy is affinity tuning: reducing the affinity of the CD3-binding arm of the TCE can temper the speed and intensity of T-cell activation, as attempted with Amgen’s AMG 340 [81]. Although AMG 340 was discontinued for other reasons, the concept remains scientifically valid and can be applied in future studies. Another approach is to use prodrug TCEs—antibodies that are activated only in the TME (for example, through tumor-specific protease cleavage)—which can localize T-cell activation to the tumor and spare systemic immune activation [82,83]. Additionally, step-up dosing regimens (gradually increasing the dose with initial low “priming” doses) have already been adopted in trials to safely reach efficacious dose levels [19,39,67]. On the immunogenicity front, fully human or humanized protein scaffolds are now the norm for minimizing ADA formation [84]. The use of continuous infusion in early trials has largely been supplanted by half-life extended formats, which are not only more convenient but also may reduce immunogenic reactions by avoiding high concentration peaks and certain administration routes (experience from the pasotuxizumab study suggested that subcutaneous delivery led to high ADA rates [8,39]). Ongoing research is also investigating adjunct therapies to manage toxicity, such as better prophylaxis or treatment for CRS (beyond tocilizumab and steroids) and monitoring for neurotoxic effects [17,49,85]. By the time TCEs reach Phase III, we anticipate that these safety optimizations will make outpatient administration with manageable side effects a realistic scenario.

### 6.6. Combination Strategies

Combining TCEs with other therapeutic modalities is a promising strategy to enhance antitumor immunity and potentially overcome the immunosuppressive TME characteristic of advanced PCa. Various combinations aimed at improving TCE efficacy are currently under clinical investigation. For instance, trials are exploring the co-administration of PSMA TCEs with CD28 (Cluster of Differentiation 28) co-stimulatory bispecific antibodies [86] or PD-1 blocking antibodies [21], seeking to provide dual signaling for enhanced T-cell activation and persistence. Another trial evaluated combining the anti-PD-1 antibody pembrolizumab with CB307, a bispecific antibody targeting PSMA on tumor cells and the 4-1BB co-stimulatory receptor on T-cells (designed to provide tumor-localized 4-1BB agonism), to augment T-cell cytotoxic activity [87].

Beyond these ongoing trials, several experimental combination approaches hold significant interest. Immune modulators, such as IL-15 super agonists, combined with TCEs could potentially stimulate robust T-cell proliferation and function [88]. Concurrently using ICIs targeting distinct pathways (e.g., anti-PD-1/L1 or anti-cytotoxic T-lymphocyte-associated protein 4 (CTLA-4)) alongside TCEs may counteract T-cell exhaustion and prolong cytotoxic activity within the TME [89]. Targeting immunosuppressive cytokines like transforming growth factor beta (TGF-β), often elevated in PCa, represents another approach to improve T-cell infiltration and enhance TCE efficacy, particularly in resistant lesions [90].

Integrating TCEs with traditional therapies is also being explored. Chemotherapy or RLT administered prior to TCEs could reduce tumor burden, potentially lessening the severity of CRS and exposing tumor antigens to prime additional immune responses [91]. Hormonal therapies, central to PCa management, might synergize with TCEs; ADT, for instance, can upregulate PSMA expression, potentially enhancing the binding and efficacy of PSMA-targeted TCEs [92]. Localized interventions like radiation therapy could induce immunogenic cell death and potentially abscopal effects, creating a TME more favorable for TCE-mediated immunity [93].

An intriguing strategy involves sequential or concurrent administration of multiple TCEs targeting different antigens [89]. Given the heterogeneity of antigen expression within tumors and over time, using TCEs against distinct targets (e.g., PSMA and STEAP1) could maximize tumor cell killing and mitigate the risk of antigen-loss escape [5,94]. While complex regarding safety and dosing, careful management could potentially amplify the “serial killing” capacity of engaged T-cells. Overall, these diverse combination strategies aim to overcome barriers such as T-cell exhaustion and TME immunosuppression, ultimately seeking deeper and more durable therapeutic responses than achievable with TCE monotherapy alone. Continued research into optimally integrated treatment regimens is essential for advancing immunotherapy outcomes in advanced PCa.

### 6.7. Advancing Clinical Development and Approval

Promising results from early-phase trials are now catalyzing larger studies. Notably, a Phase III trial of the STEAP1-targeted TCE AMG 509 is planned to confirm potential survival benefits and impact on quality of life in patients with mCRPC [95]. Success in such Phase III trials could lead to the first TCE approval for the treatment of PCa. Concurrently, other agents are progressing; for example, Regeneron’s PSMAxCD3 bispecific REGN4336 is in Phase I/II evaluation (alone and with cemiplimab) to assess its safety and preliminary efficacy [39]. Several new PSMA TCEs (e.g., JNJ-80038114) and the KLK2-directed TCE (JNJ-78278343) are currently in early clinical trials [96,97,98]. These trials will determine the optimal constructs and dosing strategies and expand our understanding of how to integrate TCEs into the treatment sequence. Over the next few years, we expect to see the first randomized trials comparing TCEs versus standard care for PCa, which will be pivotal in positioning these drugs in clinical practice.

### 6.8. Earlier Use of TCEs in the Disease Course

A significant question is whether TCEs should be moved into earlier lines of therapy rather than being reserved for late-stage mCRPC. It is logical to hypothesize that using these agents when the patient’s immune system is less compromised and tumor burden is lower could potentially increase efficacy. A Phase I trial is currently testing a PSMA-targeted TCE in patients with biochemical recurrence (rising PSA after local therapy but no visible metastases) [28,99]. This represents a setting with minimal residual disease, where immunotherapy like TCEs could theoretically eliminate micrometastatic cancer. If safe and effective, this strategy could prevent progression to overt metastases. Similarly, future trials may integrate TCEs into the upfront management of metastatic hormone-sensitive PCa (e.g., adding a TCE to ADT and AR inhibitors in high-risk patients) to determine whether it can deliver deeper responses. Moving immunotherapy earlier has been beneficial in some other cancers [100]; whether this holds true for PCa will be answered by these studies. Challenges remain; for instance, using TCEs in asymptomatic patients could alter the risk–benefit calculus, but the potential rewards are high. Researchers will also investigate whether earlier use mitigates resistance mechanisms (e.g., an untreated tumor might express more target antigen and possess a less immunosuppressive milieu than heavily treated tumors).

### 6.9. Understanding and Overcoming TCE Resistance

Addressing the critical challenge of treatment resistance, observed clinically and anticipated mechanistically, requires understanding escape pathways and developing counterstrategies. Finally, a critical research priority is to understand how tumors might resist or escape TCE therapy and how to counteract these mechanisms [101]. As previously mentioned, antigen loss or downregulation is a clear route of resistance [101]. Detailed molecular studies of post-TCE tumor samples (when available) can reveal whether resistant lesions have lower target expression or mutations affecting the target epitope. This reinforces the need for multi-targeted approaches and sequential antigen targeting. Additionally, the immunosuppressive TME rich in TGF-β, regulatory T-cells, and myeloid-derived suppressor cells in PCa could blunt TCE efficacy over time [102]. Therefore, interventions to reprogram the TME are a priority. As discussed (Section 6.6), blocking TGF-β signaling is one approach; others include using myeloid cell inhibitors (such as colony-stimulating factor 1 receptor (CSF-1R) blockers to reduce suppressive macrophages) or adding IL-12 or other cytokine therapies to boost T helper 1 (Th1) responses in the tumor. Moreover, repeated exposure to TCEs can potentially induce T-cell exhaustion. Monitoring T-cell phenotypes in patients receiving TCE therapy will inform this; high expression of exhaustion markers (PD-1, lymphocyte-activation gene 3 (LAG-3), and T-cell immunoglobulin and mucin-domain containing-3 (TIM-3)) in T-cells would support the addition of checkpoint inhibitors or switching strategies at that juncture [40]. Researchers are also exploring gene expression profiling of tumors during TCE therapy to determine how the tumor reacts; for example, determining whether interferon-stimulated genes are upregulated (indicating an immune attack) or whether escape pathways (such as alternative immune checkpoints) are induced [103]. Such data can guide rational combinations or sequence adaptations (e.g., if PD-L1 is induced in tumors, a PD-L1 inhibitor may be added). The battle between TCEs and tumor defenses is expected to evolve, and research is geared toward staying one step ahead by anticipating resistance and designing therapeutic countermeasures. In summary, the future of TCEs for PCa will be shaped by ongoing clinical trials and innovative research focusing on improving their efficacy, safety, and patient selection. The field is moving toward more targets, smarter designs, combination regimens, and earlier interventions. These efforts aim to solidify the role of TCEs as durable, safe, and widely applicable treatments for advanced PCa.

## 7. Conclusions

Having explored the fundamental mechanisms, clinical evolution, safety considerations, comparative value, and future research priorities of T-cell engagers in PCa, we return to the central premise: TCEs represent a paradigm shift in the treatment of advanced PCa, offering a means to harness the immune system against a disease that has historically been immunotherapy resistant. Our review traced the arc from initial proof-of-concept and early challenges to the increasingly sophisticated agents demonstrating tangible clinical activity today. By bridging T-cells to tumor cells, TCEs can induce tumor regression and PSA decline, even in mCRPC patients who have exhausted conventional therapies. Clinical data, although primarily from early phases, have demonstrated that prostate tumors, including aggressive variants, can be effectively targeted via T-cell redirection—a remarkable development where ICIs and vaccines alone have largely fallen short.

The strength of TCEs lies in their novel mechanisms and engineering versatility. However, the journey thus far underscores that these are potent agents requiring careful management of challenges like CRS and antigen escape through continued refinement of drug design and clinical protocols. Balanced optimism permeates the field: the potential to transform treatment is clear but requires validation and further improvements in efficacy, durability, and safety. Ongoing efforts focusing on rational combinations, biomarker-driven patient selection, and moving TCEs earlier in the disease course are crucial. While this review provides a comprehensive synthesis based on current evidence, as a narrative review, it reflects the authors’ perspective and selection of the literature; further systematic analyses may provide complementary insights.

In conclusion, T-cell engagers are poised to play a significant role in the evolving landscape of PCa therapy. They fill a critical immunotherapeutic gap for this predominantly ‘cold’ tumor, offering a flexible, off-the-shelf approach. As research continues to sharpen patient selection, fine-tune molecular designs, and integrate supportive combination therapies, TCEs hold the potential to yield more durable remissions and improve long-term control of advanced PCa. The coming years, marked by pivotal clinical trials, will be decisive. If successful, T-cell engagers will not merely add another tool but mark a new chapter, offering substantial hope for improved outcomes in a disease that remains a leading cause of cancer mortality in men. The convergence of immunology and oncology embodied by TCEs holds tremendous promise, and sustained, strategic innovation will determine how effectively we translate that promise into meaningful reality for patients.

## Figures and Tables

**Figure 1 cancers-17-01820-f001:**
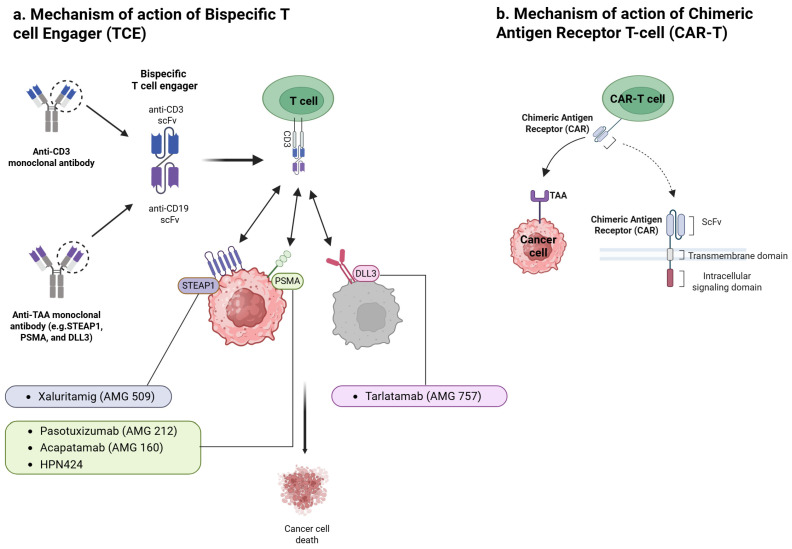
Mechanisms of action for T-cell-engaging cancer immunotherapies. (**a**) Mechanism of action of Bispecific T-cell Engager (TCE). TCEs are engineered proteins, commonly bispecific antibodies, that physically link a T-cell to a cancer cell. One arm of the TCE (e.g., an anti-CD3 scFv) binds to the CD3 complex on T-cells, while the other arm (e.g., an anti-TAA scFv) binds to a tumor-associated antigen (TAA) expressed on the cancer cell surface. This forced interaction creates an immunological synapse, leading to potent T-cell activation and targeted, MHC-independent lysis of the cancer cell. BiTE^®^ (Bispecific T-cell Engager) molecules are a specific format of scFv-based TCEs, as illustrated. In prostate cancer, relevant TAAs include prostate-specific membrane antigen (PSMA), six transmembrane epithelial antigen of the prostate 1 (STEAP1), and delta-like ligand 3 (DLL3). The figure illustrates this concept with examples of TCEs developed for prostate cancer: Xaluritamig (AMG 509) targeting STEAP1; Pasotuxizumab (AMG 212), Acapatamab (AMG 160), and HPN424 targeting PSMA; and Tarlatamab (AMG 757) targeting DLL3, all inducing cancer cell death. (**b**) Mechanism of action of chimeric antigen receptor T-cell (CAR-T). CAR-T cells are T-cells that have been genetically modified to express a synthetic Chimeric Antigen Receptor (CAR) on their surface. The CAR typically consists of an extracellular antigen-binding domain (commonly a single-chain variable fragment—scFv) specific for a TAA on cancer cells, a transmembrane domain, and one or more intracellular signaling domains. This engineered receptor allows the CAR-T cell to directly recognize and bind to the TAA on cancer cells, independently of MHC presentation. Upon antigen engagement, the intracellular signaling domains activate the CAR-T cell, leading to targeted cytotoxic killing of the cancer cell. Created with BioRender.com. Abbreviations: APC, antigen-presenting Cell; BiTE, bispecific T-cell Engager; CAR, chimeric antigen receptor; CAR-T cell, chimeric antigen receptor T-cell; CD, cluster of differentiation; CTLA-4, cytotoxic T-lymphocyte-associated protein 4; ICI, immune checkpoint inhibitor; MHC, major histocompatibility complex; PD-1, programmed cell death protein 1; PD-L1, programmed death-ligand 1; PD-L2, programmed death-ligand 2; scFv, single-chain variable fragment; TAA, tumor-associated antigen; TCE, T-cell engager.

**Figure 2 cancers-17-01820-f002:**
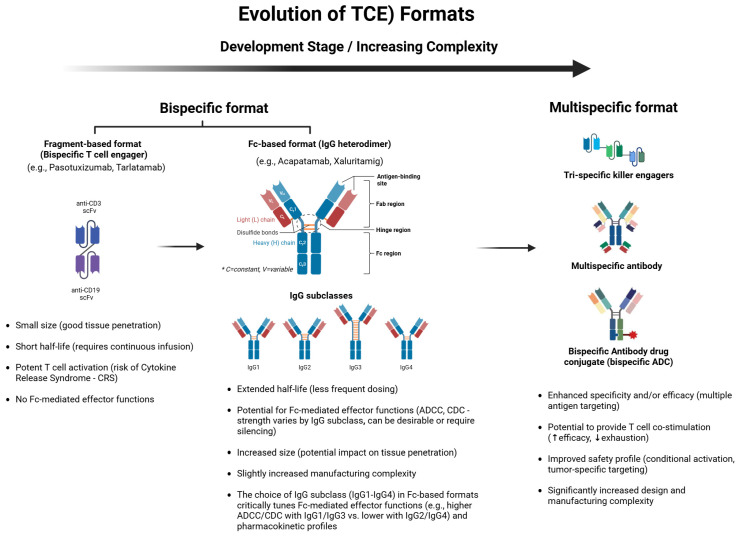
Evolution of TCE formats.

**Figure 3 cancers-17-01820-f003:**
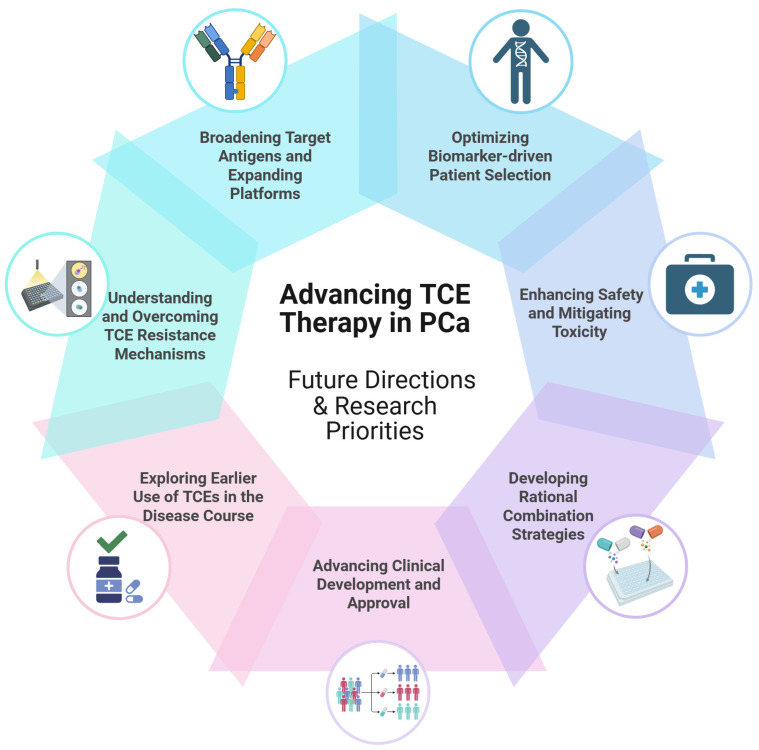
Future directions and research priorities for advancing TCE therapy in PCa. The continued development and optimization of TCE therapy for PCa encompass several key research priorities. These include broadening the range of targetable antigens and expanding TCE therapeutic platforms; optimizing biomarker-driven patient selection to identify individuals most likely to benefit; enhancing the safety profile and mitigating treatment-related toxicities; developing rational combination strategies with other therapeutic modalities; advancing clinical development pathways and streamlining regulatory approval processes; exploring the utility of TCEs in earlier stages of the disease course; and understanding and overcoming mechanisms of resistance to TCE therapy. Each area represents a critical avenue for research to maximize the therapeutic potential of TCEs in PCa. Created with BioRender.com. Abbreviations: PCa, prostate cancer; TCE, T-cell engager.

**Table 1 cancers-17-01820-t001:** Major TCE agents targeting PSMA, STEAP1, and DLL3 in clinical trials.

TCE Name, Developer	Target Antigen	Format/Key Features	Clinical Stage (Phase)	Key Efficacy Results	Key Toxicities	References
Pasotuximab (AMG 212), Amgen	PSMA	BiTE (small Fc-free)	Phase I (completed)	>50% PSA decline in some pts; Limited ORR and durability	Frequent CRS (various grades), immunogenicity	[8]
Acapatamab (AMG 160), Amgen	PSMA	IgG-like, extended half-life	Phase I (Development Discontinued)	PSA50 ~63% (evaluable pts, higher doses); modest ORR reported	CRS common (~high %), mostly G1-2 w/mitigation; Low rate G ≥ 3	[12]
HPN424, Harpoon Therapeutics	PSMA	Trispecific (PSMA × CD3 × albumin)	Phase I (Development Discontinued)	~20% had any PSA decline; ~5–6% PSA50; SD in ~50%	Mostly G1-2 CRS with step-up dosing	[21]
CC-1 (Fully Human)	PSMA	Fully human bispecific (CD3 × PSMA)	Phase I (Reported)	PSA decline up to 60% in all 14 pts (early data)	CRS common (~79%), mostly G1-2; No severe events reported	[28]
Xaluritamig (AMG 509), Amgen	STEAP1	IgG-based TCE	Phase I (Reported; Ph III Planned)	PSA50 ~49% (overall), 59% (high dose); ORR 24% (overall), 41% (high dose)	CRS ~72% (mostly G1-2, 2 G3 cases); Fatigue, myalgia; Manageable immunogenicity	[19]
Tarlatamab (AMG 757), Amgen	DLL3	BiTE-like (CD3 × DLL3)	Phase I (NEPC Cohort Reported)	NEPC cohort: ORR 10.5% (overall), 22.2% (DLL3+ subset); one durable response >2 yrs	CRS 75% (mostly G1-2, one G ≥ 3); Neurotoxicity ~12.5% (one G3)	[10]
NJ-78278343, Janssen	KLK2 (Kallikrein-2)	Bispecific antibody (CD3 × KLK2)	Phase I/II (ongoing)	Early data pending	Data pending reporting	[29]
LAVA-1207, LAVA Therapeutics	PSMA	Vγ9Vδ2 T-cell engager	Phase I/II (Ongoing)	Early signals of stable disease reported; data maturing	Generally mild AEs reported; Low CRS incidence	[30]

**Table 2 cancers-17-01820-t002:** Comparison of TCE therapy with key existing treatments for advanced prostate cancer.

Therapy	Mechanism	Typical Efficacy in Late-Line mCRPC	Key Toxicities	Potential Synergy or Considerations	References
AR-Targeted (e.g., enzalutamide, abiraterone)	Block androgen receptor signaling; efficacy loss in AR-independent disease	PSA50 ~30–40% (earlier lines); less effective later	Fatigue, metabolic disturbances (e.g., hypertension w/abiraterone)	May upregulate PSMA expression, potentially enhancing TCE activity	[54,55]
Chemotherapy (e.g., docetaxel, cabazitaxel)	Direct cytotoxicity to dividing cells (non-tumor-specific)	~35% PSA50, ORR ~10–20% (heavily pretreated); OS benefit	Myelosuppression, neuropathy, alopecia	Cytoreduction prior to TCE might lower tumor burden and reduce CRS risk	[1]
Radioligand (^177^Lu-PSMA-617)	Delivers targeted radiation to PSMA-expressing cells	PSA50 ~46%, ORR ~30% (VISION trial); survival benefit	Hematologic toxicity, xerostomia, potential renal impact	Requires PSMA+; sequential/combination w/TCE under investigation	[6,34,47,56]
Immune Checkpoint Inhibitors (e.g., pembrolizumab)	Blocks PD-1/PD-L1 or CTLA-4 to restore T-cell function	Low response (<5–10% ORR) in unselected mCRPC; higher in MSI-H/dMMR/CDK12mut	Immune-related AEs (pneumonitis, hepatitis, colitis)	May reduce T-cell exhaustion, thus enhancing TCE efficacy	[1,14]
Sipuleucel-T (autologous vaccine)	Autologous APC vaccine targeting prostatic acid phosphatase	Modest OS gain (~4 mo); low PSA50 rate	Infusion reactions, flu-like symptoms	Could be combined or sequenced with TCE for potential additive immune activation	[4,57]
T-cell Engagers (e.g., PSMAxCD3, STEAP1xCD3)	Redirect T-cells to tumor antigens (CD3 × TAA), MHC-independent	Phase I data: PSA50 ~20–60%, ORR ~10–40%+ (agent/target dependent)	Cytokine release syndrome (CRS), neurotoxicity (ICANS), potential off-target effects	Novel MOA bypassing resistance; combinations under study	[5,6,12,19]

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
