# Peer review of "T-Cell Engager Therapy in Prostate Cancer: Molecular Insights into a New Frontier in Immunotherapy"

_cancers, 2025, doi:10.3390/cancers17111820_

Round 1

Reviewer 1 Report

Comments and Suggestions for Authors
  1. Figure 1. Figure labels are way too small and the font size needs to be increased. More importantly, figures 1a and 1c appear partially redundant and essentially illustrate the same thing: how a BiTE recruits a T cell to kill a cancer cell. I suggest that a clearer distinction be made between TCEs in general and a very specific format, BiTE (trademark by Micromet), which is based on scFvs only (as is illustrated on figs 1a and 1c), unlike “bulkier” formats, as in xaluritamig or glofitamab (probably it would be a good idea to show the former on 1a, unless the authors decide to remove 1a altogether). All I am saying is that TCEs are a broad class of drugs, which include BiTEs among others, and that these terms are not interchangeable.
  2. Whereas the authors did a great job by blending in the immune checkpoint inhibitors and illustrating why ICIs have had limited success in the context of PCa, to me this appears as a deviation from the topic of the review, both as applied to the main text and to the Figure 1b. The major mechanism of action of ICIs, as presented, is a little oversimplified and does not include ICI effects on Tregs. In order to not complicate the presentation further, I suggest removing the ICI story from the figure 1, at least.
  3. Whereas there’s nothing wrong with Figure 3, to me it essentially duplicates the message outlined in Figure 1 – the MoA for BiTEs. If one substitutes the “TAA” on Figure 1 for PSMA, STEAP1 or DLL3, the essence will be retained. It is probably a good idea to delete Figure 3 and to add the specific targets/TCEs to Figure 1 to make it more informative.
  4. Line 304. Given that CC1 appears more promising compared to its predecessors, it would be really great to present its structure (if known) and provide some insight why it may work better than earlier TCEs. Actually, showing the “cartoon” structures of the lead drugs (to visually complement Table 2) being discussed in this review wouldn’t hurt at all.
  5. Line 329. It is somewhat unclear how differences in internalization kinetics may play a role in superior activity of xaluritamig (as it has to stay on cancer cell surface to be bound to the T cell). Please elaborate.
  6. Line 334. Treatment discontinuation due to CRS was indeed low, however it was still quite significant due to other TAEs. I suggest including this information.
  7. Table 2. It appears that clinical development of AMG-160 has been discontinued by Amgen. It is advisable to include this information in order to not mislead the readers. Same for HPN424.
  8. It may be worth adding a section exploring the perspectives and challenges of using BiTE-secreting T/B/liver cells, given the advent of efficient AAV- and LNP-mediated delivery approaches.
  9.  It is a little surprising that both authors share exactly the same generic email: "e-mail@e-mail.com" (lines 5 and 7).
  10. Reference 52 (line 441) does not seem to be appropriate, as it describes preclinical studies, rather than clinical observations where ADAs have resulted in diminished efficacy.

Reviewer 2 Report

Comments and Suggestions for Authors

In this narrative review the authors present current literature on T-cell Engager Therapy in Prostate Cancer.  The review covers current T cell engager therapies, mechanisms of action and evolution, different PCa targets antigens and the future directions for the field. 

The review is generally well structured and includes relevant references. There are sufficient recent citations included in the text and no obvious omissions of relevant citations.

The inclusion of comparisons between existing therapies is very useful. Additionally, the future directions section is excellent.   

The figures are clear, relevant, and help the reader to understand the concepts described.

Figure 1 concept is good but it is not clear what the difference between a) and c). The BiTE and the TCE seem to be the same. I don’t think you need the BiTe in the CAR part of the figure.

Throughout the manuscript the difference between a TCE and a BiTE is unclear, add more clarification.

Line 55: add more info on what Vaccine sipuleucel-T is

Line 147:  do the TCEs really cause the recruitment of T cells to the tumor site?

Line 215: need to expand on what is meant by knob-into-hole"

236: why are these TriTEs more stable?, expand.

I section 3 where targets are discussed add more information on what kind of receptor they are, if they have any role in cancer cell function, and expression levels in patients and over time. Also explain the co-expression of any other the targets.

303-306: how is this targeted towards gamma delta T cells, don’t both alpha beta and gamma delta express CD3?

351: explain what half life extended BiTE is and how it is different

448: expand on what these constructs are and how they work

Within section 5 add subheadings in bold and also add more clinical info on response rates and the patient populations

Section 6, some points are mentioned in multiple sections eg NK cell engagers. Remove repeated information or rearrange

555: expand on these concepts

Section 6.2

TCEs dont require MHC so in this regard NK cells aren’t really an advantage. add in what differences there are in the mechanism of action and why NK cells may be beneficial

577 explain the costim concept further

597-602. not clear what a hemi body is and section requires further explanation.

Section 6.5 : Can add further explanation on what causes the CRS

658: what is meant by a 4-1BB agonist TCE?
